# DNA polymerase theta suppresses mitotic crossing over

**Juan Carvajal-Garcia**[1], **K. Nicole Crown**[2], **Dale A. Ramsden**[1,3,4], **Jeff Sekelsky**[1,3,5]*

**1** Curriculum in Genetics and Molecular Biology, University of North Carolina, Chapel Hill, North Carolina, United States of America, **2** Department of Biology, Case Western Reserve University, Cleveland, Ohio, United States of America, **3** Lineberger Comprehensive Cancer Center, University of North Carolina, Chapel Hill, North Carolina, United States of America, **4** Department of Biochemistry and Biophysics, University of North Carolina, Chapel Hill, North Carolina, United States of America, **5** Integrative Program in Biological and Genome Sciences, University of North Carolina, Chapel Hill, North Carolina, United States of America

* sekelsky@unc.edu

**Data Availability Statement:** All relevant data are within the manuscript and its Supporting Information files.

**Funding:** Grants 1R01CA222092 and 1P01CA247773 from the National Cancer Institute

## Abstract

Polymerase theta-mediated end joining (TMEJ) is a chromosome break repair pathway that is able to rescue the lethality associated with the loss of proteins involved in early steps in homologous recombination (*e.g.*, BRCA1/2). This is due to the ability of polymerase theta (Pol θ) to use resected, 3' single stranded DNA tails to repair chromosome breaks. These resected DNA tails are also the starting substrate for homologous recombination. However, it remains unknown if TMEJ can compensate for the loss of proteins involved in more downstream steps during homologous recombination. Here we show that the Holliday junction resolvases SLX4 and GEN1 are required for viability in the absence of Pol θ in *Drosophila melanogaster*, and lack of all three proteins results in high levels of apoptosis. Flies deficient in Pol θ and SLX4 are extremely sensitive to DNA damaging agents, and mammalian cells require either Pol θ or SLX4 to survive. Our results suggest that TMEJ and Holliday junction formation/resolution share a common DNA substrate, likely a homologous recombination intermediate, that when left unrepaired leads to cell death. One major consequence of Holliday junction resolution by SLX4 and GEN1 is cancer-causing loss of heterozygosity due to mitotic crossing over. We measured mitotic crossovers in flies after a Cas9-induced chromosome break, and observed that this mutagenic form of repair is increased in the absence of Pol θ. This demonstrates that TMEJ can function upstream of the Holiday junction resolvases to protect cells from loss of heterozygosity. Our work argues that Pol θ can thus compensate for the loss of the Holliday junction resolvases by using homologous recombination intermediates, suppressing mitotic crossing over and preserving the genomic stability of cells.

## Author summary

Chromosome breaks are a common threat to the stability of DNA. Mutations in genes involved in the early steps of homologous recombination (*BRCA1* and *BRCA2*), a mostly error-free chromosome break repair pathway, lead to hereditary breast cancer. Cells

awarded to DAR. Grant 1R35GM118127 from the National Institute of General Medical Sciences awarded to JS. Grant 1K99GM118826-01 from the National Institute of General Medical Sciences awarded to KNC. The funders had no role in study design, data collection and analysis, decision to publish, or preparation of the manuscript.

**Competing interests:** The authors have declared that no competing interests exist.

lacking BRCA1 and BRCA2 rely on DNA polymerase theta, a key protein for a more error-prone pathway, for survival. Using fruit flies and mammalian cells, we have shown that mutations in genes involved in later steps of homologous recombination (*SLX4* and *GEN1*) also make cells reliant on polymerase theta. Moreover, we have shown that polymerase theta acts upstream of a type of homologous recombination that is error-prone and depends on SLX4 and GEN1. This form of homologous recombination, termed Holliday junction resolution, creates mitotic crossovers, which can lead to loss of heterozygosity and cancer. Our results expand the cellular contexts that make cells depend on polymerase theta for survival, and the substrates that this protein can use to repair chromosome breaks.

## Introduction

Double-strand breaks (DSBs) are a particularly toxic form of DNA damage. DSBs are generated during common cellular processes (*e.g.*, replication, transcription), after exposure to ionizing radiation, or by specialized mechanisms such as meiosis or the development of the adaptive immune system [1]. DSBs are also essential intermediates during nuclease-dependent genome editing. Two pathways account for most DSB repair: non-homologous end joining (NHEJ), and homologous recombination (HR) [2]. In addition, polymerase theta-mediated end joining (TMEJ) has recently been identified as a third DSB repair pathway [3–5].

DNA polymerase theta (Pol θ, gene name *POLQ*) was first shown to be involved in DSB repair in *Drosophila melanogaster* (fruit fly), and this function was found to be conserved in other invertebrates, plants, and mammals [3–7]. Inactivation of TMEJ by knocking out *POLQ* orthologs has little to no effect on organismal viability in mice, zebrafish, *Drosophila*, or *Caenorhabditis elegans*. Only when exposed to exogenous DNA damaging agents does Pol θ deficiency negatively impact survival, although to a lesser extent than when other DSB repair pathways are compromised [8–11]. However, Pol θ is required in the absence of factors that promote both NHEJ (*e.g.*, KU70 and 53BP1) [12,13] and HR (*e.g.*, BRCA1 and BRCA2) [13–15], showing that TMEJ can compensate for their loss. This is of particular interest in the context of HR-deficient breast and ovarian cancer, where Pol θ has been proposed as a promising therapeutic target [16].

HR is a multi-stage process that can lead to different repair outcomes, some of which can be detrimental [17]. An important example of detrimental HR is mitotic crossing over, as it can result in loss of heterozygosity, which can lead to cancer development [18,19]. The first step in HR is DNA end resection, which generates 3'-ended ssDNA tails. One tail is used to invade another duplex DNA molecule, forming a displacement loop (D-loop) and priming DNA synthesis. Unwinding of the D-loop and reannealing to the other end of the broken molecule completes synthesis-dependent strand annealing (SDSA). Alternatively, the D-loop may progress to form a joint molecule, the double-Holliday junction, that needs to be dissolved or resolved through cleavage for the chromosomes to be segregated; the latter process can lead to a mitotic crossover [2].

Mechanistically, how Pol θ compensates for the loss of HR proteins is largely unknown. Mutations in genes involved in early stages of HR have been shown to be synthetic lethal with *POLQ* mutations. This suggests that when these steps are inactivated, the resulting 3' ssDNA can be used by Pol θ to repair the DSB. It remains unclear whether mutations in genes involved in later steps in HR (*e.g.*, downstream of *BRCA1/2*) can similarly generate recombination intermediates that are toxic for cells in the absence of Pol θ activity.

Here we describe a strong genetic interaction between *POLQ* and the Holliday junction resolvase genes *SLX4* and *GEN1*, which encode some of the latest acting HR proteins, both in *Drosophila melanogaster* and in mammalian cells. We also show that Pol θ suppresses mitotic crossing-over in flies, thus protecting cells from this potentially pathogenic form of repair. Moreover our results, together with the observation that *POLQ* mutations have no effect in SDSA in *Drosophila* [3], argue that Pol θ is surprisingly important in processing HR intermediates even after D-loop formation.

## Results

### *Brca2* and *POLQ* mutations are synthetic lethal in *Drosophila* melanogaster

During repair of double-strand breaks (DSBs) in mammals, TMEJ is able to compensate for some HR deficiencies (Fig 1A). This is best illustrated by the requirement of *POLQ* for the survival of *BRCA1/2* mutant cancer cell lines [14,15], and the upregulation of *POLQ* in *BRCA1/2* deficient breast and ovarian tumors [14,20,21]. We therefore initially assessed whether a comparable phenomenon is evident at a whole animal level in *Drosophila*, by crossing flies heterozygous for mutations in *PolQ* and *Brca2* (the *Drosophila melanogaster* orthologs of *POLQ* and *BRCA2*; hereafter, the human gene/protein names will be used for simplicity) (Fig 1B). Homozygous mutant flies are easily identified due to the presence of a homologous balancer chromosome (*CyO*, *Cy¹ dp^{lvl} pr¹ cn²* on the second chromosome and *TM6B*, *Antp^{Hu} Tb¹ e¹ ca¹* on the third chromosome) that carries a dominant phenotypic marker (*Curly* (*Cy*) for *BRCA2*, *Humeral* (*Antp^{Hu}*) for *POLQ*) (Fig 1B). When we looked at the progeny of these flies, we observed that single mutant flies in either gene alone displayed approximately 100% viability (Fig 1C). Conversely, only 12% of the expected double homozygous mutant flies eclosed as adults, indicating semi-lethality when these two proteins are absent (Fig 1C).

Previous investigations have emphasized the strong genetic interaction between *POLQ* and genes involved in early steps of HR (*i.e.*, steps preceding D-loop formation) (Fig 1A) [13–15]. However, DNA intermediates formed downstream of end resection and strand invasion may also be amenable to repair by TMEJ. This has recently been suggested to be the case when long-range resection is impaired due to mutations in *BRCA1*, which may inhibit re-annealing of the unwound D-loop [22]. If so, mutations in genes involved in later steps of HR might also be synthetic lethal with *POLQ* mutations. Therefore, we assessed whether a genetic interaction exists between *POLQ* and genes encoding proteins involved in late steps of HR.

### Pol θ is required for viability in the absence of the Holliday junction resolvases

We decided to use *Drosophila melanogaster* to investigate the genetic relationship between Pol θ and some of the latest acting HR proteins, the Holliday junction resolvases Mus312 (SLX4 in humans), and Gen (GEN1 in humans). Human SLX4 is a scaffolding protein that coordinates at least three endonucleases: SLX1, XPF-ERCC1, and MUS81-EME1 (the interaction with MUS81-EME1 has only been reported in mammals), forming the SMX tri-nuclease [23–27]. GEN1 acts independently of SLX4 [28]. These structure-specific endonucleases have both unique and overlapping DNA substrate specificities [29–31].

We assessed the viability of every double mutant combination (*POLQ SLX4*, *POLQ GEN1*, and *SLX4 GEN1*) as well as the triple mutant (*POLQ SLX4 GEN1*) by crossing heterozygous flies and comparing the fraction of adult homozygous mutant flies observed to what would be expected by Mendelian genetics. While *POLQ SLX4*, *POLQ GEN1*, and *SLX4 GEN1* double mutant combinations are fully viable, flies that lack Pol θ, SLX4, and GEN1 rarely progress to

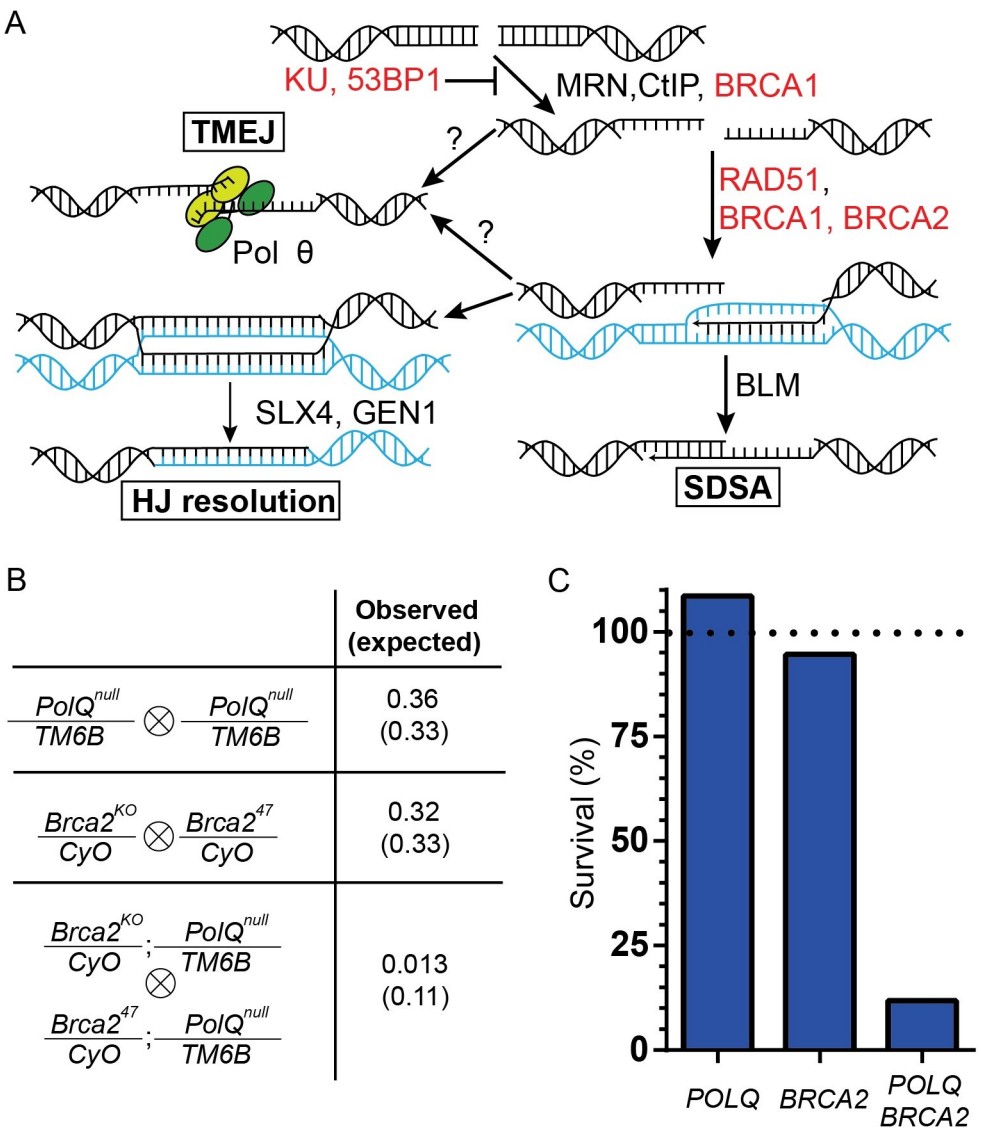

**Fig 1. The genetic interaction between *POLQ* and *BRCA2* is conserved in *Drosophila melanogaster*.** A) Schematic of the DSB pathways following end resection, including a partial list of proteins involved in each step. Synthetic lethal with Pol θ shown in red. B) Genotypes of the flies crossed to assess the viability of *PolQ*, *Brca2*, and *PolQ Brca2* mutants (left). Fraction of homozygous mutant flies observed and, in parentheses, expected by Mendelian genetics; the *Cy* and *Antp^Hu^* markers, present in *CyO* and *TM6B* respectively, are recessive lethal. C) Observed survival of homozygous mutant files for the indicated genes expressed as percent of expected. Horizontal dashed line at Y = 100 indicates 100% survival. N = 696 (*POLQ*), 331 (*BRCA2*), 612 (*POLQ BRCA2*).

adulthood (<1% survival) (Fig 2A and S1 Table). When using the *PolQ^null^* allele over *PolQ^Z2003^* (*PolQ^Z2003^* is a nonsense mutation reported to be a strong hypomorph; see Materials and methods), we observed a 3% survival for *POLQ SLX4 GEN1* mutant flies (n = 1059). This is, to our knowledge, the first evidence for synthetic lethality for *POLQ* and genes required for steps in HR after strand invasion.

These results indicate a genetic redundancy between Pol θ and the resolvases. The functions of the resolvases suggested that the synthetic lethality could be due to a role for Pol θ in rescuing unresolved HR intermediates that arise from spontaneous DSBs, or stalled or broken replication forks. If this is the case, we reasoned such roles would be apparent as sensitivity to

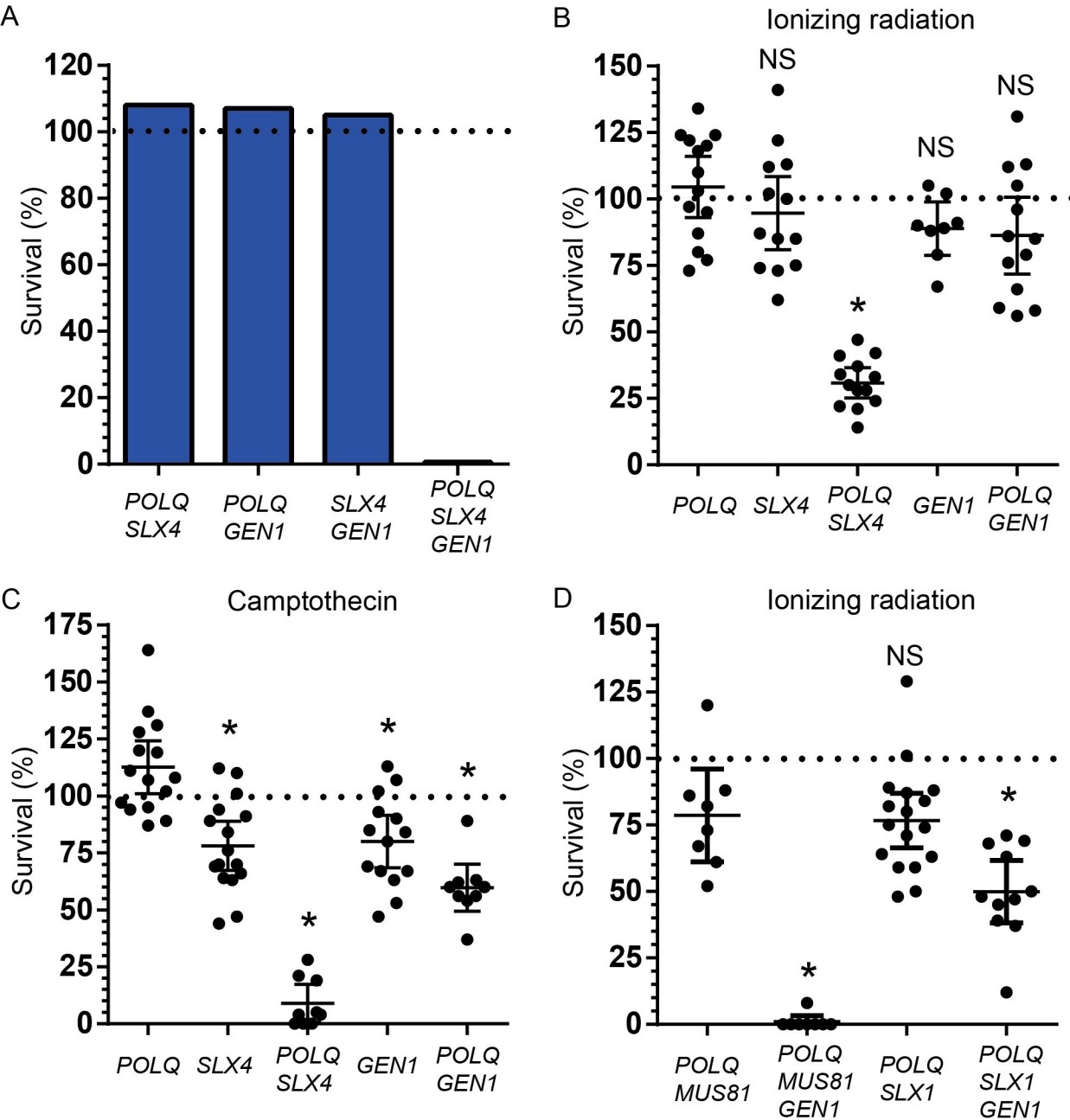

**Fig 2. *POLQ* is required for viability in the absence *SLX4* and *GEN1*.** A) Observed survival of flies homozygous mutant for the indicated genes expressed as percent of expected. n = 1126 (*POLQ SLX4*), 747 (*POLQ GEN1*), 257 (*SLX4 GEN1*), 448 (*POLQ SLX4 GEN1*). Since *POLQ*, *SLX4*, and *GEN1* are on the third chromosome, the balancer *TM6B*, *Antp^Hu Tb^1 e^1 ca^1* was used in these crosses. B), C) and D) Survival of flies exposed to 1000 rads of ionizing radiation (B and D) or 10 μM camptothecin (C) homozygous mutant for the indicated genes relative to the untreated progeny of the same parents. Each dot represents a vial pair. Horizontal dashed line at Y = 100 indicates 100% survival. Error bars represent 95% confidence intervals (CI). Statistical significance was assessed by one-way ANOVA with Bonferroni correction to account for multiple comparisons; ns, not significant; *, p <0.05.

exogenous DNA damaging agents in double mutants that are viable in the absence of such agents.

We used ionizing radiation (IR) to induce DSBs, and camptothecin, a type I topoisomerase poison, to generate stalled and broken replication forks. We compared the sensitivity of

*POLQ*, *SLX4*, and *GEN1* single mutants, as well as *POLQ SLX4*, and *POLQ GEN1* double mutant flies, to moderate doses of IR (1000 rads) and camptothecin (10 μM). All three single mutants showed an average survival of ≥80% for both DNA damaging agents (Fig 2B and 2C and S2 Table). *POLQ SLX4* double mutant flies showed the strongest reduction of viability, 31% and 9% survival when treated with IR or camptothecin, respectively (Fig 2B and 2C and S2 Table). *POLQ GEN1* double mutants showed only a modest reduction in viability. Pol θ is thus more important for cell viability in the absence of SLX4 than in the absence of GEN1. These results show that DSBs and collapsed or broken replication forks generate DNA substrates, likely HR intermediates, that require the use of Pol θ or SLX4 for repair.

We also tested whether SLX1 or MUS81, two of the nucleases that associate with SLX4, played a more significant role than the other in the repair of these intermediates. We observed mild sensitivity to IR of both *POLQ MUS81* and *POLQ SLX1* double mutants (Fig 2D and S2 Table), reflecting an apparent redundancy between these two nucleases in the presence of SLX4 and GEN1. Interestingly, *POLQ MUS81 GEN1* triple mutant files are much more sensitive to IR (1% survival) than *POLQ SLX1 GEN1* triple mutant flies (50% survival) (Fig 2D and S2 Table), which suggests that MUS81 is required for the repair of certain DNA substrates in the absence of GEN1.

Next, we addressed whether this genetic interaction observed in flies is conserved in mammals. For this, we used T-antigen transformed mouse embryonic fibroblasts (MEFs) derived from isogenic wild type (wt) and *Polq*[-/-] mice [8]. In addition, we used *Polq*[-/-] MEFs that have been complemented with the human *POLQ* cDNA [5]. We electroporated ribo-nucleoprotein complexes of purified *Staphylococcus pyogenes* Cas9 protein with gRNAs targeting either the non-protein-coding *Rosa26* locus (control locus, *R26*) or exon 4 in *SLX4* (Fig 3A). 72 hours later, we assayed cell viability by a colony formation assay. In addition, we harvested DNA from the cells, amplified the genomic regions across the Cas9 site and used tracking of indels by decomposition (TIDE) [32] to calculate the fraction of chromatids that had an indel at the target sites (% editing) (Fig 3A). Targeting *SLX4* did not decrease viability in wt or in complemented *Polq*[-/-] MEFs compared to targeting the non-coding locus (Figs 3B and S1 and S3 Table). However, we observed a 54% reduction in viability in the *Polq*[-/-] MEFs when targeting *SLX4*, relative to the control locus, which matches the editing efficiency of 58% in that cell line (Figs 3B and S1 and S3 Table). Unlike flies, this decrease in viability in *POLQ SLX4* double mutants MEFs is observed in the absence of exogenous DNA damage (except for the DSB made by Cas9), arguing the genetic interaction between *POLQ* and *SLX4* is stronger in mammalian cells than it is in flies.

## Lack of Pol θ and resolvases leads to high levels of apoptosis

Interestingly, etched tergites (disrupted tissue patterning in the abdomen) could be readily observed in most *POLQ SLX4* double mutant flies (88.1%, n = 42) (Fig 4A). These are indicative of defects in cell survival or proliferation during development. We never observed them in wt (n = 71) and *POLQ* mutants (n = 40) and rarely in the *SLX4* ones (18.2%, n = 44). This phenomenon has been described in *POLQ RAD51* double mutants [3].

To accurately quantify the level of apoptosis in flies with different genotypes, we used an antibody that detects cleaved Dcp-1, a marker of apoptosis in *Drosophila* [33]. We immunostained larval wing imaginal discs, a highly proliferative tissue that becomes the adult wings after metamorphosis. The use of a larval tissue also allows us to assess the levels of apoptosis in *POLQ SLX4 GEN1* flies, at least in the fraction of animals that reach the larval stage. We observed very little apoptosis in *POLQ* mutant flies, while levels of apoptosis were significantly higher in *POLQ SLX4*, and even higher in the *POLQ SLX4 GEN1* triple mutant (Fig 4B and 4C

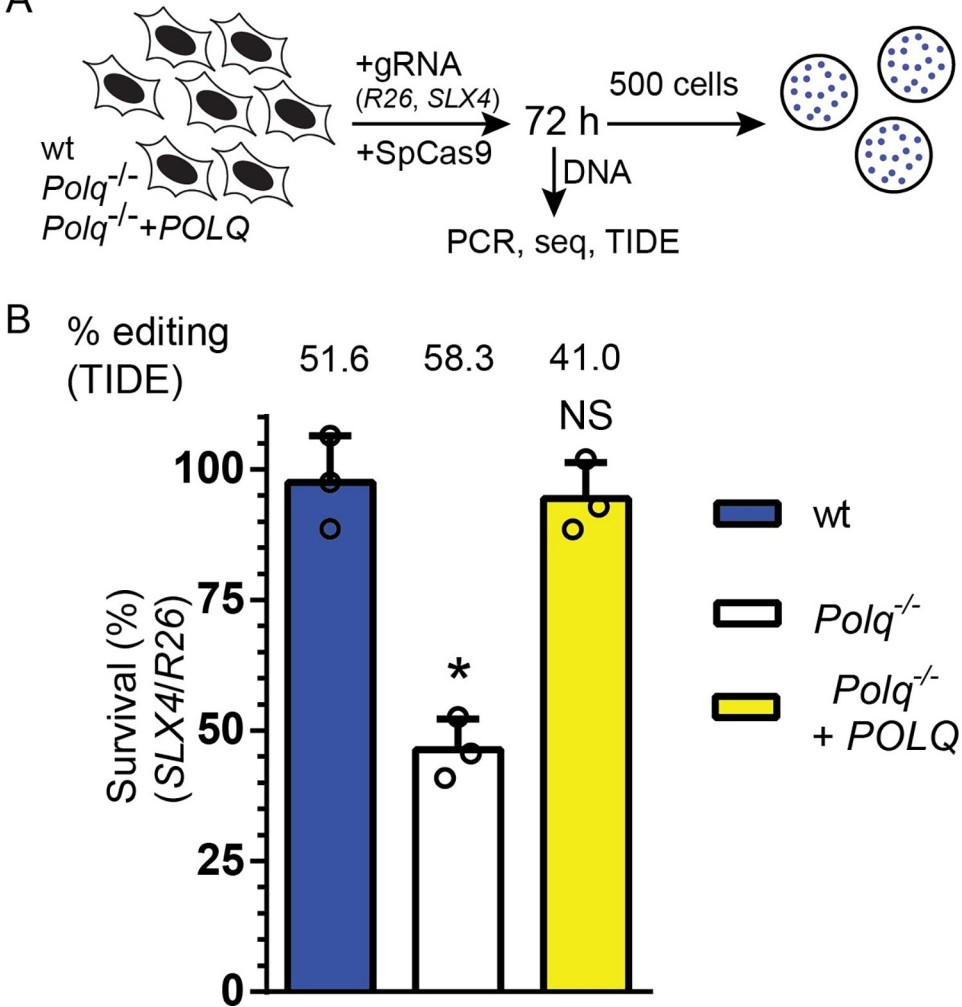

**Fig 3. Genetic interaction between POLQ and SLX4 in mouse cell culture.** A) wt, *Polq⁻/⁻* and complemented *Polq⁻/⁻*
MEFs were electroporated with Cas9 targeted with a gRNA to the *Rosa26* (R26) locus or to *Slx4*. 72 hours later, 500
cells were plated into each of three plates to assay viability. Genomic DNA from the remaining cells was used as a
template for amplification around the breaks. PCR product was sequenced and editing efficiency was calculated with
TIDE. B) Survival after Cas9 cleavage targeted by the *SLX4* gRNA, relative to the *R26* gRNA for each cell line. Editing
efficiency is indicated above the graph. Error bars represent standard error of the mean, n = 3 biological replicates.
Statistical significance was assessed by one-way ANOVA with Bonferroni correction to account for multiple
comparisons; ns, not significant; *, p <0.05.

and S4 Table). This is consistent with the reduction in viability observed in the *POLQ SLX4
GEN1* triple mutant, as well as sensitivity to exogenous DNA damage by IR or camptothecin in
the *POLQ SLX4* double mutant.

## Pol θ suppresses mitotic crossovers

The strong genetic interaction between Pol θ and the resolvases suggests the existence of a
DNA intermediate that will either be joined by TMEJ or progress to a double Holliday junc-
tion and be resolved by SLX4 or GEN1. This DNA intermediate, when left unrepaired, causes
cell death. We hypothesize that this substrate is an HR intermediate. In *Drosophila* somatic
cells, both TMEJ and Holliday junction formation are downstream of the preferred HR

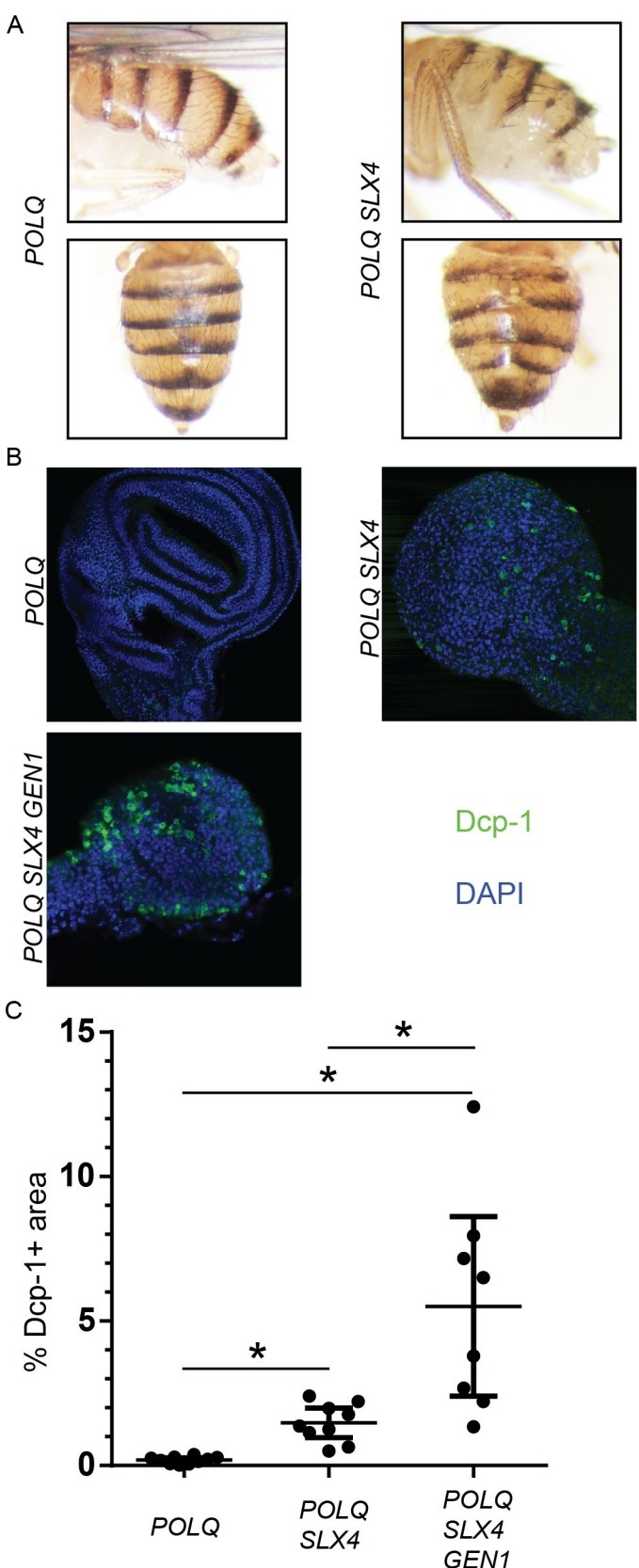

Dcp-1

DAPI

**Fig 4. POLQ SLX4 GEN1 flies have high levels of apoptosis.** A) Images of representative *POLQ* mutant and *POLQ SLX4* mutant female flies. For the bottom pictures, the wings and legs were removed. B) Images of representative wing imaginal discs from third instar larvae of the indicated genotypes stained with an anti-Dcp1 antibody (green) and DAPI (blue). C) Quantification the Dcp-1 signal expressed as the percent of the area of Dcp-1 within each disc. Error bars represent 95% CI. Statistical significance was assessed by one-way ANOVA with Bonferroni correction to account for multiple comparisons; *, p <0.05.

pathway, SDSA. Support for this hypothesis comes from the finding that Pol θ-dependent end joining products and mitotic crossovers are both increased when SDSA is inactive due to the absence of the BLM helicase [34,35]. This leads to a model in which DNA intermediates formed after aborted SDSA can then be processed by either TMEJ or the structure-specific endonucleases. In the absence of both pathways, these DNA intermediates accumulate and become toxic to cells, which ultimately undergo apoptosis; high levels of apoptosis lead to organismal death.

We set out to identify potential consequences of the epistatic relationship between TMEJ and Holliday junction resolution described above by designing a DSB repair assay in *Drosophila* that allows for assessment of an expected product of Holliday junction resolution, mitotic crossovers (Fig 5A). DSBs are generated in the germline cells of male flies by expressing Cas9 under a germline promoter (*nos*), and a gRNA, expressed with the *U6* promoter, targeting the coding region of the *rosy* (*ry*) gene, located in the right arm of chromosome 3. Homozygous *ry* mutant flies are viable and have an easily identifiable mutant eye color. Only the maternal chromosome gets cut, as the paternal allele harbors a SNP that alters the protospacer adjacent motif (PAM) sequence (TGG becomes TGA) required for recognition and cleavage by Cas9 (Fig 5A).

This assay allows us to detect mutagenic end joining, homologous recombination events that used the homologous chromosome as a template, and unedited (never cut or precisely repaired) chromosomes. Moreover, we can characterize HR events as crossovers or non-crossovers due to the presence of the phenotypic markers *scarlet* (*st*) and *ebony* (*e*), as well as the fact that *Drosophila* males don't generate crossovers during meiosis [35].

We performed this assay using 60 single males, six of which were sterile. We randomly selected one progeny fly from each of the 54 remaining males, and detected editing in 40 (74%), showing that the assay is highly efficient (Fig 5B). In wild-type flies we observed that repair of a DSB by end joining (EJ) and HR are roughly equally common (EJ: 21/54, 39%; HR: 19/54, 35%) (Fig 5B).

Mitotic crossovers are present in only 0.2% of wild type flies (Fig 5C and 5D and S5 Table); strikingly, they are present at 18-fold higher levels in *POLQ* deficient flies (Fig 5D and S5 Table). Interestingly, ablation of all resolvase activity (*i.e.*, both SLX4 and GEN1) was required to completely eliminate mitotic crossing over. This is in contrast to mitotic crossovers generated in the absence of the anti-crossover helicase FANCM, which depend solely on SLX4 [36], and are likely not originated by a blunt DSB like the ones in this assay.

Because *nos* is expressed early in the male germline, it should be noted that repair events might be amplified unevenly during cell proliferation prior to spermatogenesis. Even though we don't expect this to disproportionately affect different genotypes, we analyzed these results in a different way by assessing only whether each male had some crossover progeny or no crossover progeny. The results of this analysis mirrored those in the previous one, though the magnitude of the change was lower (3.5X more mitotic crossovers in *POLQ* mutant flies than in wt flies) (Table 1). This latter analysis is definitively unaffected by unequal expansion, but presumably underestimates the amount of crossing over due to our inability to distinguish between one and multiple crossover events in the same male germline.

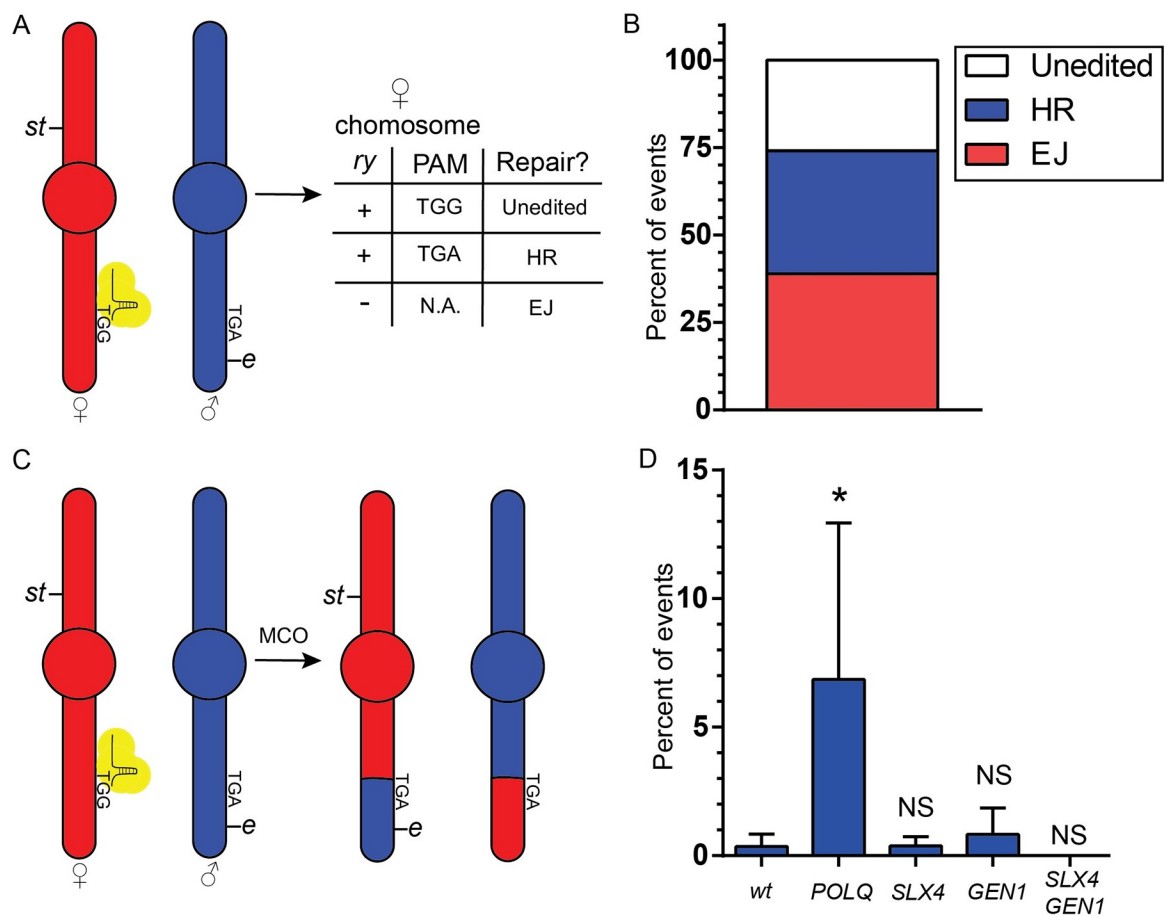

**Fig 5. Pol θ suppresses mitotic crossing over.** A) Third chromosomes (maternal, ♀, and paternal, ♂) of a male fly expressing Cas9 and a gRNA targeting the *ry* gene, and heterozygous for the markers *st* and *e*. The PAM sequence for the gRNA in the maternal chromosome, and lack thereof in the paternal one, is indicated. Wild-type *ry* (+) in the maternal chromosome indicates either unedited (if PAM is present) or HR repair (if PAM is absent). Mutant *ry* (-) indicates repair by mutagenic end joining. B) Percent of chromosomes repaired by HR, EJ, or unedited, calculated by assessing the *ry* and PAM status. n = 56. C) Third chromosomes of a male fly described in panel A before and after the generation of a mitotic crossover. D) Percent of total progeny that were recombinant is indicated for each genotype (like those described in panels A and C). Males were maternally and zygotically mutant for the indicated genes. Number of males crossed: 97 (wild type), 55 (*POLQ*), 62 (*SLX4*), 60 (*GEN1*), 69 (*SLX4 GEN1*). Number of progeny scored: 8790 (wild type), 3471 (*POLQ*), 5016 (*SLX4*), 6830 (*GEN1*), 3446 (*SLX4, GEN1*). Error bars represent 95% CI. Statistical significance was assessed by one-way ANOVA with Bonferroni correction to account for multiple comparisons; ns, not significant; *, p <0.05.

These results show that the absence of Pol θ increases the amount of mitotic crossing over during HR. Moreover, our results imply that Pol θ can act upstream of the Holliday junction resolvases, and thus presumably upstream of Holliday junction formation as well.

**Table 1. Mitotic crossing over is increased in *POLQ* mutant flies.** Crosses from for Fig 5D were characterized as having any progeny with a mitotic crossing over (Yes) or no progeny with a mitotic crossover (No). *p*-value for each mutant genotype when compared to wt was calculated with a $X^2$ test applying the Yates correction.

| Genotype | MCOs | | $p$ ($X^2$ with Yate's correction) |
|---|---|---|---|
| | **Yes** | **No** | |
| wt | 5 | 92 | N.A. |
| *POLQ* | 10 | 45 | 0.021 |
| *SLX4* | 7 | 55 | 0.26 |
| *GEN1* | 6 | 54 | 0.40 |
| *SLX4 GEN1* | 0 | 69 | 0.15 |

## Discussion

Pol θ has the ability to compensate for the loss of BRCA1 and BRCA2, key mediators of HR, as well as for loss of proteins involved in NHEJ [12,14,15]. Moreover, a recent synthetic lethality screen uncovered 140 genes that have a synthetic growth defect with *POLQ*, most of which operate outside of DSB repair, and showed that as much as 30% of breast tumors may be relying on Pol θ for survival [13]. This ability has motivated the search for a Pol θ inhibitor for treatment of cancer [37].

However, no HR gene outside of the resection/strand invasion step has been shown to be synthetic lethal with *POLQ*. Here we show that flies deficient in Pol θ, SLX4, and GEN1 –the latter of two acting late during HR–are inviable, due to high levels of apoptosis likely caused by endogenous DNA damage, and that flies with mutations in *POLQ* and *SLX4* are hypertensive to the DNA damaging agents IR and camptothecin. Moreover, we demonstrate that the genetic interaction between Pol θ and SLX4 is conserved in mice. This striking genetic redundancy strongly suggests that TMEJ and Holliday junction formation/resolution are involved in processing similar DNA substrates.

The ability of Pol θ to rescue deficiencies in HR genes is not completely understood. A well-defined starting substrate for TMEJ is generated after 5' resection of both ends of a DSB [5,12], yet it is not known whether that is the only substrate used by Pol θ. Two 3' ssDNA tails are also the starting substrate in HR, implying a possible competition between TMEJ and HR. The difficulty in accurately measuring the different outcomes of HR in mammalian cells has led to conflicting evidence on whether Pol θ has the ability to suppress HR, and therefore compete for a starting substrate [12,14,15].

Well characterized assays in *Drosophila* allow for the unambiguous assessment of SDSA, the major pathway for completion of repair by HR in somatic cells [34], and they show that lack of Pol θ doesn't affect the frequency of DSB-induced SDSA [3]. Pol θ deficiency similarly doesn't affect the frequency of single strand annealing, another pathway immediately downstream of end resection, in flies or in human cells [38,39]. This argues that Pol θ does not compete for the 3' ends generated by 5' end-resection.

In contrast, Pol θ suppresses mitotic crossovers and is synthetic lethal with resolvase deficiency, arguing it does compete for repair by the alternate means for completion of HR that involves a double Holliday junction. SDSA is upstream of TMEJ and Holliday junction formation/resolution, yet both Pol θ-associated indels and mitotic crossovers are observed in wild-type flies. This indicates that sometimes SDSA either fails or cannot be completed. We propose that the remaining DNA intermediate(s) can either be joined by Pol θ, generating a small indel, or can progress to a double Holliday junction, that may be resolved to create a mitotic crossover.

Thus, though the generation of small indels is implicit to repair by TMEJ, this pathway protects against potentially more deleterious forms of repair, such as larger deletions [21], or inter-homolog recombination after a DSB is made in both homologs [40]. Holliday junction resolution also generates genotypes, in the form of loss of heterozygosity, that can affect whole chromosome arms. The high potential pathogenicity of these events may make them more detrimental to cells than small indels, supporting Pol θ's role in maintaining genomic stability.

## Materials and methods

### Drosophila stocks

*Drosophila* stocks were kept at 25°C on standard cornmeal media (Archon Scientific). Mutant alleles were obtained from the Bloomington Drosophila Stock Center (BDSC) or were a gift

from Dr. Mitch McVey and have been described in [41] (*Brca2$^{KO}$*), [42] (*Brca2$^{47}$*), [43] (*PolQ$^{null}$*) and [3] (*PolQ$^{Z2003}$*), [44] (*mus312$^{D1}$* and *mus312$^{Z1973}$*), [45] (*Gen$^{Z5997}$*, *slx1$^{F931}$* and *slx1$^{e01051}$*), and [46] (*mus81$^{Nhe}$*). *PolQ$^{null}$* (a deletion) was used either homozygous (Figs 1, 2 and 4), or *in trans* to *PolQ$^{Z2003}$*, a nonsense mutation reported to be severely hypomorphic [3] (Fig 5). *Brca2* and *mus312* alleles were used compound heterozygous. *Gen$^{Z5997}$* was used hemizygous over the deficiency *Df(3L)6103*. Since *mus81* is in the X chromosome, *mus81$^{Nhe}$* was used homozygous in females and hemizygous in males. Allele-specific PCR was used to detect the presence of the mutant alleles in recombinant chromosomes (primers in S6 Table).

Pictures of fly abdomens shown in Fig 4A were taken with a Swiftcam 16 Megapixel Camera, and the Swift Imaging 3.0 software.

Flies expressing *Streptococcus pyogenes* Cas9 controlled by the *nanos* promoter, inserted on the X chromosome (attPA2) were obtained from BDSC (stock number 54591 [47]).

Flies expressing a gRNA targeting the *rosy* (*ry*) locus (5'-CATTGTGGCGGAGATCTC GA-3') were generated by cloning the gRNA sequence into the pCFD3 plasmid (Addgene #49410) as in [47]. The gRNA construct was stably integrated into an *attP* landing site at 58A using phi-C31 targeting (stock number 24484) (Best Gene).

For the generation of flies with a deletion of the *ry* locus, two gRNA sequences were cloned into the pU6-BbsI-chiRNA plasmid (Addgene #45946) [48]. One gRNA targeted 5' of the *ry* start site (5'-GGCCATGTCTAGGGGTTACG-3') and the other targeted 3' of the *ry* stop codon (5'-GATATGCACAGAATGCGCCT-3'). These were injected along with the pHsp70-Cas9 plasmid (Addgene #45945) [48] into a *w$^{1118}$* stock (Best Gene). The resulting *ry* deletion starts 373 bp upstream of the *ry* start codon and ends 1048 bp downstream of the *ry* stop codon.

## DNA damage survival assays

Survival in the presence of DNA damaging agents was determined as in [49]. Five females and three males carrying heterozygous mutations for the indicated genes were allowed to mate and to lay eggs for 72 hours (untreated progeny), when they were moved to a new vial where they laid for 48 hours (treated progeny). The latter brood was exposed to 1000 rads of ionizing radiation (source: $^{137}$Cs) or 10 μM camptocethin, diluted from a concentrated stock in a 10% ethanol, 2% Polysorbate 20 aqueous solution. The fraction of heteroallelic mutant flies in the treated progeny was divided by the fraction of heteroallelic mutant flies in the untreated progeny to calculate the survival.

## Statistical analysis

Experiments that employ statistical tests as indicated in the figure legends were done using GraphPad Prism 6 (ANOVA) or Excel ($X^2$ test).

## Cell lines

Mouse Embryonic Fibroblasts (MEFs) were made from isogenic wt or *Polq*-null mice generated by conventional knock-out [8] that were obtained from Jackson Laboratories and maintained on a C57BL/6J background and immortalized with T antigen as described in [5]. Cells were incubated at 37°C, 5% $CO_2$ and cultured in DMEM (Gibco) with 10% Fetal Bovine Serum (VWR Life Science Seradigm) and Penicillin (5 U/ml, Sigma). All lines used in this study were certified to be free of mycoplasma by a qPCR [50] with a detection limit below 10 genomes/ml. In addition, cell lines were randomly selected for third party validation using Hoechst staining [51].

## Clonogenic survival assay

Transfections were performed as in [21]. Genome targeting ribonucleotide-protein complexes (RNP) were made by annealing the indicated crRNA (*R26*: 5'-ACTCCAGTCTTTCTAGAA GA-3', *SLX4*: 5'-ACAGCAGGAGTTTAGAAGGG-3') to a tracrRNA (Alt-R, IDT) to form 8.4 pmol of gRNA, followed by incubation of annealed gRNA with 7 pmol of purified Cas9 (made after expression of Addgene #69090) [52]. The assembled RNPs were electroporated into 200,000 MEFs along with 32ng of pMAX-GFP using the Neon system (Invitrogen) in a 10 ul tip with one 1,350 V, 30 ms pulse and plated (three electroporations formed one biological replicate). After 72 h, 500 cells were plated into 3 different plates and let grow for 7 days to allow for colonies to form. Cells were fixed and stained as in [53], using a 6% glutaraldehyde, 0.5% crystal violet aqueous solution. Colonies were counted and survival was calculated for each cell line individually. Genomic DNA for the remaining cells was harvested and used as a template for the generation of a PCR product surrounding the *R26* or the *SLX4* break site (primers in S6 Table). This PCR product was sequenced (Eton) and the editing efficiency was calculated using TIDE [32]. The editing efficiencies for the SLX4 break site are noted in the figure; editing efficiencies for the *R26* break site were 84.7%, 95.7% and 95.3% for wt, *Polq*$^{-/-}$ and *Polq*$^{-/-}$ + *POLQ* respectively.

## Wing imaginal disc immunofluorescence

The anterior halves of third instar larvae of third instar, 5-7-day old, homozygous mutant for the indicated genes, larvae were dissected in phosphate-buffered saline (PBS), everted, and fixed in 4% formaldehyde at room temperature for 45 min. They were washed three times in PBS+0.1% Triton-X (PBSTx), blocked in 5% normal goat serum for one hour at room temperature, and incubated overnight at 4˚C in a 1:100 dilution of cleaved Dcp-1 antibody (Cell signaling #9578S) in PBSTx. Larva heads were then washed six times with PBSTx and incubated in a 1:500 dilution of secondary antibody (goat anti-Rabbit IgG, Alexa Fluor 488, Life Technologies) for two hours at room temperature. After washing six times in PBSTx, DAPI was added at a 1:1000 dilution. Discs were dissected and mounted in 50 ul of Fluoromount G mounting media (Thermo).

Pictures were taken with a Zeiss LSM880 confocal laser scanning microscope using a 40X oil immersion objective with a constant gain and a 0.6X zoom using ZEN software. Images were saved as.czi files and were processed and the signal was quantified using ImageJ as in [54].

## Mitotic crossover assay

For Fig 4B, single males expressing Cas9 and the gRNA targeting the *ry* gene were generated (see cross below).

$$nos :: Cas9; \frac{st}{TM6B} \; x \; \frac{U6 :: gRNA}{CyO}; \frac{e}{TM6B} \rightarrow \frac{Cas9}{Y}; \frac{U6 :: gRNA}{+}; \frac{st}{e}$$

In addition, these males were heterozygous for $st^1$ and $e^1$ as well as for a SNP that changes the PAM sequence recognized by Cas9 immediately downstream of the gRNA sequence in *ry* (the chromosome with the mutation in *st* has the functional PAM and will be cut by Cas9). These males were crossed to females that were $e^1$ over TM6B, *Antp*$^{Hu}$ $Tb^1$ $e^1$ $ca^1$. To characterize the repair event that occurred after the DSB, a single male progeny, heterozygous for *e* and *Antp*$^{Hu}$, was crossed to females homozygous for a deletion in *ry*. If the non- *Antp*$^{Hu}$ progeny has rosy eye color, the repair event was characterized as mutagenic end joining (EJ). If the

non- $Antp^{Hu}$ progeny had wild-type eye color, genomic DNA from a single male was extracted and the DNA surrounding the break was amplified by PCR (primers in S6 Table). The presence of the silent mutation that changes the PAM sequence, revealed by resistance to cutting by *Bcc*I of the PCR product surrounding the Cas9 target site, was interpreted as HR. The presence of the intact PAM was characterized as unedited.

For Fig 4D and Table 1, single males as the ones described above and with maternal and zygotic mutations in the indicated genes (see crosses used to generate them below), where crossed to flies homozygous mutant for *st* and *e*.

$$wt : nos :: Cas9; \frac{stPolQ^{Z2003}}{TM6B} x \frac{U6 :: gRNA}{CyO}; \frac{e}{TM6B} \rightarrow \frac{Cas9}{Y}; \frac{U6 :: gRNA}{+}; \frac{stPolQ^{Z2003}}{e}$$

$$POLQ : nos :: Cas9; \frac{st\ PolQ^{Z2003}}{PolQ^{null}e} x \frac{U6 :: gRNA}{CyO}; \frac{PolQ^{null}e}{TM6B} \rightarrow \frac{Cas9}{Y}; \frac{U6 :: gRNA}{+}; \frac{st\ PolQ^{Z2003}}{PolQ^{null}e}$$

$$SLX4 : nos :: Cas9; \frac{mus312^{Z1973}st}{mus312^{D1}e} x \frac{U6 :: gRNA}{CyO}; \frac{mus312^{D1}e}{TM6B} \rightarrow \frac{Cas9}{Y}; \frac{U6 :: gRNA}{+}; \frac{mus312^{Z1973}st}{mus312^{D1}e}$$

$$GEN1 : nos :: Cas9; \frac{Gen^{Z5997}st}{Df(3L)6103e} x \frac{U6 :: gRNA}{CyO}; \frac{Df(3L)6103e}{TM6B} \rightarrow \frac{Cas9}{Y}; \frac{U6 :: gRNA}{+}; \frac{gen^{Z5997}st}{Df(3L)6103e}$$

$$SLX4\ GEN1 : nos :: Cas9; \frac{Gen^{Z5997}\ mus312^{Z1973}st}{Df(3L)6103\ mus312^{D1}e} x \frac{U6 :: gRNA}{CyO}; \frac{Df(3L)6103\ mus312^{D1}e}{TM6B}$$

$$\rightarrow \frac{Cas9}{Y}; \frac{U6 :: gRNA}{+}; \frac{Gen^{Z5997}\ mus312^{Z1973}st}{Df(3L)6103\ mus312^{D1}e}$$

Flies that were wild type for both markers or mutant for both markers were characterized as having a crossover event.

## Supporting information

**S1 Fig. Representative images of one plate per condition (genotype and gRNA) scored for Fig 3B.**
(TIF)

**S1 Table. Number of heterozygous and homozygous mutant flies scored for Fig 2A, and % of mutant flies expected and observed.**
(CSV)

**S2 Table. Number of flies heterozygous (balanced) and homozygous mutant (unbalanced), treated or untreated with the indicated mutagen, scored for Fig 2B–2D, and calculated % survival for each vial pair.**
(CSV)

**S3 Table. Number of colonies counted, for each biological replicate of cells of the indicated genotype transfected with Cas9 and the indicated gRNA, and calculated viability relative to the *R26* gRNA represented in Fig 3B.**
(CSV)

**S4 Table. Area of each wing disc in pixels and area of Dcp-1 positive signal within that disc in pixels for discs of the indicated genotype, as well as the calculated % area positive for**

**Dcp-1 represented in Fig 4C.**
(CSV)

**S5 Table. Number progeny from each male that didn't have crossover (NCO) or that did (MCO), as well as the percentage of the progeny that had a crossover, represented in Fig 5D.**
(CSV)

**S6 Table. Primers used in this study.**
(DOCX)

# Acknowledgments

The authors would like to acknowledge Dr Mitch McVey for providing the *PolQ^null^*, *PolQ^Z2003^*, *BRCA^KO^* and *BRCA^47^* flies, Dr. Rick Wood and Matthew Yousefzadeh for *Polq^-/-^* MEFs, as well as Susan McMahan for technical assistance.

# Author Contributions

**Conceptualization:** Juan Carvajal-Garcia.

**Data curation:** Juan Carvajal-Garcia.

**Formal analysis:** Juan Carvajal-Garcia.

**Funding acquisition:** K. Nicole Crown, Dale A. Ramsden, Jeff Sekelsky.

**Investigation:** Juan Carvajal-Garcia.

**Methodology:** Juan Carvajal-Garcia.

**Project administration:** Dale A. Ramsden, Jeff Sekelsky.

**Resources:** K. Nicole Crown.

**Supervision:** Dale A. Ramsden, Jeff Sekelsky.

**Visualization:** Juan Carvajal-Garcia.

**Writing – original draft:** Juan Carvajal-Garcia, Dale A. Ramsden, Jeff Sekelsky.

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
