## [Decision Letter · Decision Letter 0]

18 Dec 2020

Dear Dr Sekelsky,

Thank you very much for submitting your Research Article entitled 'DNA polymerase theta suppresses mitotic crossing over' to PLOS Genetics.

The manuscript was fully evaluated at the editorial level and by independent peer reviewers. The reviewers appreciated the attention to an important problem, but raised some substantial concerns about the current manuscript. Based on the reviews, we will not be able to accept this version of the manuscript, but we would be willing to review a much-revised version. We cannot, of course, promise publication at that time.

In your revision, please pay close attention to the additional experimental details and primary/raw data presentation issues raised by Reviewer 2. In addition, Reviewer 3 has raised an important point about POL theta affecting other homologous recombination (HR) steps, such as end resection. While End-Seq may not be feasible for this study, particularly in flies, the authors are encouraged to consider ways to address it in their experimental system, such as measuring gene conversion tracks in their fly allelic HR experiments, which could also be a consequence of deregulated end resection. In addition, editing the text to discuss all plausible scenarios for how POL theta is influencing HR is recommended.

If you decide to revise the manuscript for further consideration at PLOS Genetics, please aim to resubmit within the next 60 days, unless it will take extra time to address the concerns of the reviewers, in which case we would appreciate an expected resubmission date by email to plosgenetics@plos.org.

[LINK]

We are sorry that we cannot be more positive about your manuscript at this stage. Please do not hesitate to contact us if you have any concerns or questions.

Yours sincerely,

Jeremy M. Stark

Guest Editor

PLOS Genetics

David Kwiatkowski

Section Editor: Cancer Genetics

PLOS Genetics

Reviewer's Responses to Questions

**Comments to the Authors:**

Reviewer #1: This manuscripts requires very little commenting by me. The abstract provides a accurate summary of the data presented and the novelty of the study: extending our knowledge on the synthetic lethal interaction between Pol theta on the on hand, and homologous recombination on the other. The paper is very concise but makes the point on shared intermediates between TMEJ and Holliday junction formation/resolution quite clearly and convincingly by a set of genetic experiments making use of an elaborate set of single, double and triple mutant flies . The manuscript is well written in the sense that it was easy to read (not always the situation for Drosophila papers). It is also appreciated that the genetic interaction as validated in mammalian cells. The notion that TMEJ suppresses mitotic crossing over, hence preserving genomic stability (at the expense of indels) is interesting.

I have only two small comments:

i) in some instances, the wording "share a common substrate" may give the impression an incorrect impression, while I realise it is not what the authors suggest. For instance, the scenario that TMEJ can process unresolved HR intermediates that may manifest in SLX4 mutants does not mean that HR intermediates are intrinsic to TMEJ (page 9 line 189/190). I realise that the sentence could be read such that the common DNA substrate is DSBs with 3' ssDNA tails but it is nevertheless a bit confusing.

ii) The idea that TMEJ can act on HR intermediates post D-loop invasion was recently reported in worms (Kamp et al., Nat. commun. 2020). Include this study as it seems highly relevant.

Reviewer #2: This is an interesting investigation into the role of polymerase theta in DNA repair and its relationship with other proteins involved in homologous and nonhomologous repair of DNA double strand breaks. The results would be a useful addition to the literature except for, primarily, one significant problem. That problem relates to data presentation, and the fact that, as presented, the results are presented only after the authors have analyzed them as they wish - the raw data are nowhere to be found. This makes it quite difficult to judge the validity of the authors' conclusions, and it does not allow readers (or reviewers) to consider alternative interpretations. This deficiency must be remedied. The presentation of results would be fine for a seminar, but not for a publication.

Detailed critiques below:

line 108 - should be Humeral (not Humoral). This marker, and Curly, are presumably on balancers. Please specify which chromosomes are being used here, and throughout. In all cases the actual crosses should be given. This is an even more significant issue in later experiments.

line 127 - present detailed results please, not just a summary. In the double mutant cross, were single mutant viabilities scored, and did they confirm results from previous two crosses? Or was there possibly some dominant interaction?

line 171 - Expression of Cas9 is causing damage - so this experiment is not done "in the absence of exogenous damage". It would be useful to confirm this result by another method, perhaps using RNAi to knock down SLX4.

line 174 and experiment that follows - the cuticular defect mentioned is properly referred to as "etched tergites". Unfortunately, the photo is a very poor representation of what, I imagine, the authors observed. First, the photos have an extreme yellow tint, making it difficult to see the cuticle; second, the arrow seems to rest right on top of the disturbed cuticle; third, there is no quantitation of this defect, making it impossible to judge its significance. Since this study examines apoptosis in wings, were there any wing patterning defects?

line 185 (and 246) - the authors state that the viability reduction of the POLQ SLX4 Gen1 triple mutant is "due to endogenous DNA damage". This has certainly not been demonstrated. It seems likely that this is true, but it has not been shown that these proteins have no roles outside of DNA repair. The authors should slightly temper this statement.

line 198 (experiment described starting here) - this experiment definitely needs a more complete and detailed description. What were the precise crosses? Did Cas9 come maternally or paternally? (Since its driven by nanos, this matters.) What were maternal and paternal genotypes with respect to mutants being analyzed? These details must be shown - the results cannot be fully evaluated without them.

line 219 - "was required to" completely "eliminate mitotic crossing over" (add for clarity)

line 220 - "is" in "contrast to..." (grammar)

Table 1 - needs a legend explaining details of experiment

lines 300, 301 - use of term "heteroallelic mutant flies" in line 300 and "homozygous mutant flies" on line 301, in the same sentence, is confusing. I believe the authors are referring to the same genotype in both cases.

line 326 - "Heads of third instar larvae..." Do larvae have heads?

line 327 - "inverted"? Please elaborate.

line 340 - "these flies were heterozygous" for "the genes..." (grammar)

line 352 - "single males as the ones described above..." Which males?? Clarify please. Showing crosses would help with this.

Figure 2 - Each dot represents a vial pair? Please give more detail in legend or in Methods. Also, this method of presenting results, as a ratio of treated to untreated, obscures the underlying data. The actual results obtained in the experiment should be presented, not just this abstract graphical treatment. Also, which differences are significant?

Figure 3 - Reporting only relative survival obscures absolute survival numbers (i.e., overall impact of cutting, replating). Does Cas9 cutting affect survival of cells, particularly polq mutant cells?

Figure 4 - part A needs a better photo (and quantitation); part B - these discs do not appear to be at the same stage of development - could that impact the results?

Figure 5 - This experiment really needs to show the crosses used. In part A - were both chromosomes ry+ to start? I believe this to be the case, but the Figure makes it confusing because it labels the mutant alleles (st and e) in exactly the same way as the wildtype allele (ry, no +). Part D - It is not entirely clear what we're looking at here: is this % of progeny that are recombinants, or % of fathers that produced any recombinants? Neither the explanation in the legend, or in the text, or in Methods is clear enough to know the answer to this with certainty. I also have other questions: Were reciprocal crossovers recovered equally? Were crossovers ever associated with the occurrence of ry mutants? If so, which recombinant were they on? Finally, how many males were tested? (Answer in Table 1?) How many progeny were scored?

In sum, this paper appears to represent significant and interesting work. When the flawed presentation of data is remedied, it should be reconsidered.

Reviewer #3: This is paper from the Sekelsky Lab reports a synthetic lethality between POLQ and the HJ resolvases SLX4 and GEN1 in fruit fly. While it was previously known that TMEJ repair via POLQ is necessary for survival of BRCA1/2 mutant cells, a synthetic genetic interaction with genes involved in later steps of HR was not previously demonstrated. The authors expand their finding to mammals and show that in mouse embryonic fibroblasts, the lack of POLQ and SLX4 result in a strong genetic interaction. Back to flies, the authors observe drastic increase in apoptosis in cells lacking POLQ, SLX4 and GEN1. Of importance, POLQ depletion in flies leads to 18-fold increase in mitotic crossovers after Cas9-induced chromosome break, consistent with the model that in the absence of POLQ, repair intermediates are directed to HR repair, HJ formation and the action of SLX4 and/or GEN1.

Overall, this is a well structured, well written and straightforward manuscript. The reported findings are novel and should have an important impact in the DNA repair community. The major point of improvement is related to the proposed model for how the lack of POLQ results in the increased requirement for resolvase action. The authors don't provide a clear model and don't elaborate much on this key part of the paper. The authors should consider the possibility that altered resection dynamics upon lack of POLQ may be triggering the increase in HJ formation (see below). In addition, the results in the paper also beg the question as the whether POLQ is also required upon lack of BLM.

Major points

1. The authors mention that "TMEJ and Holliday junction formation/resolution share a common substrate” but fail to clarify/discuss which substrate this may be. The authors should consider that changes in DNA end resection in the absence of POLQ may be causing the requirement for the resolvases. In the absence of POLQ, DNA ends with short resected tracts, which would mostly be repaired by TMEJ, may have more opportunity to be further ressected, leading to accumulation of ends with longer resected tracks that may become the substrate for canonical HR and the formation of more HJs. The authors mention that SDSA frequency is not altered by lack of POLQ, and seem to hastily conclude that POLQ does not compete for the 3’ ends generated by 5’ end-resection. However, this does not seem a compelling enough argument to prove that there are not more ends channeling to canonical HR and HJ formation. It would be useful if the authors are able to monitor resection (preferably using End-Seq) and assess whether more ends are resected, and longer resected tracks are generated upon lack of POLQ.

2. Taken together, the work does beg the question as to whether POLQ deletion is synthetic lethal with BLM deletion. Even if SDSA frequency is not altered by lack of POLQ, one wonders if the role of BLM in promoting dissolution would become essential in the absence of POLQ. In fact, an interesting prediction is that in the absence of BLM, POLQ could play an even more important role in limiting drastic increases in mitotic crossing overs. Exploration of the POLQ and BLM genetic interaction could significantly increase the impact of the paper and further strengthen the central claim that POLQ suppresses mitotic crossing over.

Minor points

I. On Figure 3, it would be reassuring if the authors could show a western blot confirming that the editing of the SLX4 gene impacted SLX4 protein levels, as well as show representative images for the colony formation survival assay.

II. For most presented data in the paper, including survival data from flies or mouse cells, please show p-value whenever appropriate. I understand that p-values are not always essential/relevant, but for cases where the authors think p-value are not necessary, I would like a quick explanation as for why.

III. For Figure 4A, besides the representative image, it would be useful to show the number of POLQ and SLX4 mutant female flies that presented the defects in abdominal banding pattern and to discuss a little more of how POLQ and SLX4 could be related to this developmental defect, as observed on the POLQ RAD51 flies.

**Have all data underlying the figures and results presented in the manuscript been provided?**

Reviewer #1: Yes

Reviewer #2: **No: **All given in the review above

Reviewer #3: Yes

PLOS authors have the option to publish the peer review history of their article (what does this mean?). If published, this will include your full peer review and any attached files.

Reviewer #1: No

Reviewer #2: No

Reviewer #3: No

---

## [Decision Letter · Decision Letter 1]

27 Feb 2021

Dear Dr Sekelsky,

We are pleased to inform you that your manuscript entitled "DNA polymerase theta suppresses mitotic crossing over" has been editorially accepted for publication in PLOS Genetics. Congratulations!

For the final files during the formatting review mentioned below, please consider the editorial suggestions of reviewer #2 for the legends of Figs 2 and 5.

Yours sincerely,

Jeremy M. Stark

Guest Editor

PLOS Genetics

David Kwiatkowski

Section Editor: Cancer Genetics

PLOS Genetics

Comments from the reviewers (if applicable):

Reviewer's Responses to Questions

**Comments to the Authors:**

Reviewer #2: I am satisfied that the authors have done a satisfactory job of dealing with critiques of the original manuscript.

very minor:

The Figure 2 legend has a typo (dote). Also, instead of stating that flies were homozygous for the indicated genes, it might be better to say they were homozygous mutants for the indicated genes.

Figure 5 legend is still not completely intelligible: "For each male (like the oneones described in panel A. and C),

671 percent of its progeny with a crossover."

Something like, "Percent of total progeny that were recombinant is indicated for each genotype" might be clearer.

Reviewer #3: I agree with the responses provided by the authors and support publication of the revised manuscript.

**Have all data underlying the figures and results presented in the manuscript been provided?**

Reviewer #2: Yes

Reviewer #3: Yes

PLOS authors have the option to publish the peer review history of their article (what does this mean?). If published, this will include your full peer review and any attached files.

Reviewer #2: No

Reviewer #3: No

**Data Deposition**

http://datadryad.org/submit?journalID=pgenetics&manu=PGENETICS-D-20-01774R1

**Press Queries**

---

## [Editor Report · Acceptance letter]

18 Mar 2021

PGENETICS-D-20-01774R1 

DNA polymerase theta suppresses mitotic crossing over 

Dear Dr Sekelsky, 

We are pleased to inform you that your manuscript entitled "DNA polymerase theta suppresses mitotic crossing over" has been formally accepted for publication in PLOS Genetics! Your manuscript is now with our production department and you will be notified of the publication date in due course.

With kind regards,

Katalin Szabo

PLOS Genetics

On behalf of:
